# Development of Flexible Dispense-Printed Electrochemical Immunosensor for Aflatoxin M1 Detection in Milk

**DOI:** 10.3390/s19183912

**Published:** 2019-09-11

**Authors:** Biresaw Demelash Abera, Aniello Falco, Pietro Ibba, Giuseppe Cantarella, Luisa Petti, Paolo Lugli

**Affiliations:** Faculty of Science and Technology, Free University of Bolzano-Bozen, 39100 Bolzano, Italy; Aniello.Falco@unibz.it (A.F.); pibba@unibz.it (P.I.); Giuseppe.Cantarella@unibz.it (G.C.); Paolo.Lugli@unibz.it (P.L.)

**Keywords:** aflatoxin M1, milk, dispense-printing, biosensor, immunosensor, sensitivity

## Abstract

Detection of mycotoxins, especially aflatoxin M1 (AFM1), in milk is crucial to be able to guarantee food quality and safety. In recent years, biosensors have been emerging as a fast, reliable and low-cost technique for the detection of this toxin. In this work, flexible biosensors were fabricated using dispense-printed electrodes, which were functionalized with single-walled carbon nanotubes (SWCNTs) and subsequently coated with specific antibodies to improve their sensitivity. Next, the immunosensor was tested for the detection of AFM1 in buffer solution and a spiked milk sample using a chronoamperometric technique. Results showed that the working range of the sensors was 0.01 µg/L at minimum and 1 µg/L at maximum in both buffer and spiked milk. The lower limit of detection of the SWCNT-functionalized sensor was 0.02 µg/L, which indicates an improved sensitivity compared to the sensors reported so far. The sensitivity and detection range were in accordance with the limitation values imposed by regulations on milk and its products. Therefore, considering the low fabrication cost, the ease of operation, and the rapid read-out, the use of this sensor could contribute to safeguarding consumers’ health.

## 1. Introduction

The growth of different fungal genii like *Aspergillus*, *Fusarium*, *Penicillium*, *Claviceps*, and *Alternania* under favorable conditions in a variety of food products and in animal feed leads to the synthesis of their secondary metabolites, called mycotoxins [1,2,3]. It is estimated that approximately 25% of crops worldwide are contaminated with mycotoxin [4,5]. This results in economic losses as well as human health problems, especially in the case of ingestion of contaminated food [2]. The best known mycotoxins are aflatoxins, ochratoxins, fumonisins, patulin, and trichothecenes. These toxins can represent potential hazards to human beings [1], since they are carcinogenic, teratogenic, mutagenic, nephrotoxic, hepatotoxic, immunotoxic, and may also damage the nerve system [2]. In particular, aflatoxin M1 (AFM1), a metabolic product of aflatoxin B1 (AFB1), is present in milk and milk products [6] and is the most toxic compound in this group [7,8]. When AFB1 is consumed by dairy animals through contaminated feed, it is changed into its enzymatic hydroxylated product, AFM1 [7,9,10], which is finally secreted into milk by the mammary glands [6,9]. The conversion rate of consumed AFB1 into AFM1 is estimated to be 0.3–6.2% [7,11], and appears in milk 12–24 h after ingestion [12,13]. Although AFM1 has less toxicity than its parent compound AFB1 [9,10,14], the International Agency for the Research on Cancer (IARC) initially classified it as a Group 2B human carcinogen, and recently reviewed its carcinogenicity categorization and moved it to Group 1 [11,12,15]. AFM1 is relatively thermo-stable during milk processing techniques like pasteurization, sterilization, and chilling, and also during production of dairy products [11,16]. Its amount cannot be reduced through processing, and it can therefore become a substantial risk to human health. Currently, many countries have set regulations for aflatoxins [17]. For example, the European Union has set the maximum level of AFM1 in milk to 0.025 µg/kg (ppb) and 0.05 µg/kg for infants and adults, respectively [18]. The former are more exposed to AFM1 since they are the highest consumers of milk and related products.

The detection of AFM1 in milk is therefore crucial to guaranteeing food quality and safety, and thereby safeguarding consumer health. State-of-the-art of detection of AFM1 is carried out using classical methods like chromatography and enzyme-linked immunosorbent assay (ELISA) [19], which are time-consuming and require expensive equipment and high expertise in addition to sample pre-treatment (e.g., extraction, purification, and detection and quantification) [11,12,20,21]. To overcome these limitations, it is crucial to develop a sensing method which can be fast, reliable, sensitive, and selective. In this regard, biosensors can be used to analyze toxins, combining biological receptors, (e.g., antibodies, enzymes, nucleic acids) with a physical transducer, which in turn translates the response into an electrical signal [21,22]. In this way, a real-time observation of the antibody–antigen interaction can be performed [23]. Compared to conventional methods, biosensors appear to be simple, inexpensive, portable, fast, easy to operate, and highly sensitive [24].

Nowadays, screen printing is the most widely used technique for the fabrication of biosensors [25]. However, it needs a mask to structure the biosensor, it consumes a considerably high amount of ink, and it requires a precise thickness modulation and expensive machines. To overcome such problems, other printing techniques like dispense-printing can be employed. Dispense-printing fabrication methods are mask free, low-cost, and minimize ink usage. The materials (ink) can be changed by changing a cartridge, and any design modification can be implemented during production by simply changing a CAD file. 

In this study, we fabricated biosensor electrodes through dispense-printing techniques. The working electrode was functionalized with single-walled carbon nanotube (SWCNTs) to improve its sensitivity, and then immobilized or coated with toxin-specific antibodies for selectivity. Our aim was to use this printing technique to manufacture flexible biosensors as well as to improve the sensitivity and limit of detection for AFM1 through SWCNT functionalization. To the best of our knowledge, this is the first work demonstrating dispense-printed flexible electrodes for immunosensor applications.

## 2. Materials and Methods

### 2.1. Reagents and Solution

All reagents and solutions used in this work were analytical-grade, unless otherwise specified. AFM1 (from *Aspergillus flavus*), anti-rat immune-pure antibody (raised in goat), hydrogen peroxide (H_2_O_2_), tween 20 (surfactant detergent), polyvinyl alcohol (PVA), methanol, phosphate saline buffer (PBS tablets), 3,3′,5,5′-tetramethylbenzidine dihydrochloride (TMB), and carbonate buffer (CB capsules) were purchased from Sigma Aldrich. The monoclonal antibodies (MAb) against AFM1 (mouse IgG), AFM1–HRP conjugate and reaction stopping acidic solution (from standard AFM1 ELISA kit), and citrate buffer containing 0.1M KCl were purchased from 2B Scientific (Upper Heyford, UK). Ink pastes for printing electrodes (silver ECI 1011 and silver/silver chloride ECI 6038E) were obtained from LOCTITE E&C, Irvine, CA, USA. SWCNTs were obtained from Carbon Solution Ink, CSI, Riverside, CA, USA. Polyethylene terephthalate (PET) flexible substrates with a 125 micron thickness were obtained from Mylar, USA. Commercial milk samples (for testing after spiking) were used. 

Preparation of SWCNT solution**:** SWCNT solution was prepared according to Reference [26]. 

Preparation of PBS solution: One tablet of PBS was dissolved in 200 ml of distilled water to obtain 0.01 M of PBS solution, which was equivalent to 137 mM NaCl, 2.7 mM KCl and 10 mM phosphate buffer solution. The pH of the buffer was adjusted to 7.4 at 25 °C.

PBS with Tween20 (PBS-T) was prepared by adding 0.05% (0.5 ml) of Tween20 (*v*/*v*) into PBS and was used as a surfactant detergent for washing. 

Preparation of CB solution: One capsule of CB was dissolved in 100 mL of distilled water to obtain 0.05 M carbonate–bicarbonate buffer, which was equivalent to 1.59 g of Na_2_CO_3_ and 2.93 g NaHCO_3_ in 1 L distilled water. Its pH was adjusted to 9.6 at 25 °C. All the buffer solutions were stored at 4 °C. 

AFM1 standard solutions: AFM1 stock solution was prepared by dissolving it in methanol at a concentration of 10 mg/mL, and was then kept at −18 °C. The standard solutions (0.01, 0.02, 0.05, 0.1, 0.5, and 1 µg/L) were obtained by diluting the stock solution with PBS buffer using serial dilution. Aflatoxins are subject to degradation when exposed to UV light; therefore, the standard solutions were kept in brown reagent bottles and/ or covered with aluminum foil during the experiment.

AFM1 antibody solutions: Anti-rat monoclonal antibody (MAb) solution (20 µg/mL) was diluted in 10 mM PBS buffer. 

Anti-primary antibody (IgG) solution: IgG solution was prepared with a concentration of 10 µg/mL in 50 mM CB buffer.

Spiked milk: The milk used for spiking the standard sample was AFM1 free. To assess the performance of the immunosensor, different standard solutions of AFM1 were spiked into milk. Before spiking, the milk was centrifuged at 6000 rpm for 10 min. 

### 2.2. Procedure

#### 2.2.1. Fabrication

The overall process of electrode fabrication is schematically shown in Figure 1. The electrodes were printed using a dispense printer (Voltera V-one, Canada) on a 125 µm thick flexible PET substrate. The working (WE) and counter (CE) electrodes were printed with Ag paste, while the reference electrode (RE) was printed with Ag/AgCl paste (see Figure 1A,B). After fabrication, the electrodes were cured for 15 min at 120 °C for ink hardening. SWCNTs were uniformly spray-deposited only on top of the WE using a spray deposition unit (Nordson E4 EFD, UK) (Figure 1C) and a shadow mask.

At this stage, electrical (surface resistance and cyclic voltammetry) and material analysis (morphology and thickness) was performed on the biosensor electrodes, which is presented in Section 2.2.2. After the electrode analysis, the fabrication was finalized with the antibody deposition (Figure 1D). The development of the immunosensor required the immobilization of the antibodies, the blocking of uncovered active sites, and the competition of the analyte with the conjugate. These three steps were done similarly to in References [22,27]. First, the antibodies were immobilized (directly drop casted) on the top of the WE only. This process was performed by precoating the WE with 8 µL of 10 µg/mL secondary antibody anti-IgG (raised from mouse) solution in 50 mM carbonate buffer (pH 9.6) and then kept overnight at 4 °C. The sample was then washed with 50 µL of PBS-T (0.05% Tween 20 in 10 mM PBS buffer, pH 7.4), rinsed with 18.2 MΩ distilled water, and kept at room temperature until dry. Next, the WE was coated with a blocking solution (1% PVA) to block the possible remaining active sites of the electrode and to avoid non-specific binding, and then washed with 50 µL of PBS-T. Finally, 8 µL of primary antibody (MAb) (20 µg/mL of monoclonal anti-AFM1 in PBS, raised in goat) was coated and incubated for 2 h at 37 °C and stored at 4 °C until used. The cross-reactivity rate of MAb was 100% for AFM1 and 30% for AFB1. As a last step, for the competition of the analyte with the conjugate, 4 µL of PBS buffer or spiked milk sample, both containing AFM1 standards, was placed on top of the WE with 4 µL of the AFM1–HRP conjugate, obtained from an ELISA kit (1:10 dilution). Next, curing was performed at 37 °C for 2 h. The sensor was rinsed with 50 µL of PBS-T and kept at room temperature to dry. A solution of 70 µL of 5 mM of TMB and 1 mM H_2_O_2_ in citrate buffer, that contained 0.1 M KCl, was then drop-casted onto the three electrodes (CE, WE, and RE) and kept for 25 min at room temperature. Finally, 70 µL of stopping solution (2N H_2_SO_4_) was added to stop the enzymatic reaction and the current was measured. The current values were collected for 3 min every 50 ms, using a custom-made LabVIEW 2017 software operated in chronoamperometry mode with a potential of −100 mV. 

#### 2.2.2. Characterization

Before antibody immobilization, electrical and material analysis of the electrode was carried out by performing resistance measurements, cyclic voltammetry, surface analysis, and thickness evaluation. The electrical properties of the WE were characterized in terms of surface resistance and cyclic voltammetry, using a sourcemeter (KEITHLEY 2614B SourceMeter®, a Tektronix Company, Beaverton, OR, USA). The surface resistance measurement was carried out at five different positions of the electrode and the average was taken. To measure cyclic voltammetry, phosphate buffer (100 mM, pH 7.0) was used in the presence of KCl as the electrolyte substance. The morphology of the electrodes was characterized by atomic force microscope (AFM) (CoreAFM, Nanoserf, Sweden), while the thickness was measured with a non-contact 3D-optical profilometer (ProFilm3D from Filmetrics, Unterhaching, Germany). 

#### 2.2.3. Testing

Calibration curve for immunosensor: The amperometric measurement was performed using a range of AFM1 standard solutions (0, 0.01, 0.02, 0.05, 0.1, 0.5, and 1 µg/L) prepared in PBS to plot a calibration curve. 

Analysis of milk sample*:* The milk was defatted by centrifuging at 6000 rpm for 15 min at 4 °C. The two phases, fat and cream, were discarded and the skimmed milk was collected and used for the experiment. Analysis of AFM1 in skimmed milk was carried out by spiking with known concentrations of AFM1 (0, 0.01, 0.02, 0.05, 0.1, 0.5, and 1 µg/L). 

Safety awareness: Since AFM1 is a carcinogenic metabolite, it must be managed with extreme care. All glassware that had been in contact with AFM1 was dipped in 10% sulfuric acid for overnight and then washed extensively with distilled water. 

## 3. Results and Discussion 

### 3.1. Characterization of the Electrode

Prior to immobilization, the electrical property, morphology, and thickness of the electrodes were characterized before and after functionalization with SWCNTs.

Characterization of electrical properties: The sheet resistance (R_sh_) of the WE prior to spray deposition of SWCNTs was measured with a four-point probe system, using a source meter. It was calculated according to Reference [28] and resulted 4.08 ± 0.06 Ω/□ for unfunctionalized dispense-printed electrode (n = 3), and functionalization increased the R_sh_ to 1.12 ± 0.16 kΩ/□. This increase of R_sh_ was due to the semiconducting nature of the SWCNTs. 

Figure 2 shows the cyclic voltammograms of the unfunctionalized (without SWCNTs, in black) and functionalized electrodes (with SWCNTs, in red) (n = 3) in a 5 mM potassium hexacyanoferrate (III) solution with 0.1 M KCl. 

Theoretically, the ratio of peak cathodic current (I_pc_) to peak anodic current (I_pa_) should be one, since for reversible reactions, the reduction and oxidation are assumed to be equal. In our experiment, this ratio was 1.36 for the unfunctionalized electrode, which was only slightly different from the theoretical value. On the other hand, this value was 1.43 for the functionalized electrodes. The applied potential at I_pc_ and I_pa_ was the cathodic (E_pc_) and anodic (E_pa_) peak potential, respectively. We measured a peak-to-peak separation value of 0.09 V for both electrodes. The deviation of the latter from its theoretical value was due to the conducting nature of the spray-coated SWCNTs [29,30].

Characterization of morphological properties: The adhesion properties of the inks onto the substrate were investigated by observing the edges of the electrodes using an optical microscope at different magnifications (picture not presented). A good precision of the printing method was found, allowing printing of even single separate lines, making the spreading of the ink negligible at the edges. The atomic force microscope (AFM) images of the WE are given in Figure 3. For the unfunctionalized electrode (Figure 3A), it was observed how the Ag particles were almost uniformly distributed, resulting in an improved electrical property of the electrode. Figure 3B shows the AFM picture on PET substrate with a similar procedure of spray deposition onto the electrode. Regarding the functionalized electrode, SWCNTs were uniformly distributed on the surface of the working electrode and formed a network. During spray deposition, more SWCNTs were deposited on the center of the spraying line and formed a thicker network. 

Another key material property of the electrodes was their thickness, which controlled the electrical properties. The 2D profiles for the thickness measurements and roughness are given in Figure 4, where the thickness was measured in terms of step height, while the roughness was given by root mean square (RMS, (S_q_)). A non-vertical step was obtained due to the printer settings, which caused a height difference between the central part of the electrode and its edge (see Figure 4A). The average step height of the electrode was 13.75 µm, while the surface roughness as shown in Figure 4B was 0.9838 µm.

### 3.2. Testing of the Immunosensor Performance

After fabrication, characterization, functionalization, and immobilization of antibodies, the immunosensor was tested for AFM1 sensitivity in buffer solution and to test the matrix effect of the sample in defatted skimmed milk samples. The sensor response (generated current), shown in Table 1 and Table 2, for different AFM1 concentrations depended on the antibody antigen (analyte) integration. This generated current due to the integration can be taken as evidence for the attachment of the antibody onto the WE. In this work, since TMB has superior detection properties, it was chosen to determine the activity of HRP as a mediator for H_2_O_2_. The analysis was performed using chronoamperometry at a supplied potential of −100 mV, at which the TMB product undergoes reduction [27]. AFM1–horseradish peroxidase conjugate (AFM1–HRP) was used in this immunoassay. The HRP-catalyzed oxidation of TMB resulted in an oxidized substrate, which in turn changed its color to blue. The intensity of this blue color change was inversely proportional to the concentration of AFM1. Stop solution (sulfuric acid) was then added to inactivate the enzyme (in this case HRP), stabilizing the color development and allowing an accurate measurement. The reaction stopped when the HRP was inactivated, which took 25 min at room temperature. 

#### 3.2.1. Testing in a Buffer Solution

AFM1 standards prepared in PBS solution were used to test both unfunctionalized and functionalized biosensors. The results are shown in Figure 5. The limits of detection were 0.049 µg/L and 0.02 µg/L for unfunctionalized and SWCNT-functionalized sensors, respectively. The SWCNT-functionalized immunosensor was able to detect the presence of this toxin from well below the limits imposed by the law, even for infants (0.025 µg/L). SWCNT functionalization improved sensitivity and the limit of detection. Our functionalized sensor was also more sensitive than the screen-printed biosensors previously reported by References [22,27], characterized by limits of detection of 0.025 µg/L and 0.039 µg/L, respectively. The variation of current generated due to the concentration variation of AFM1 had an inverse relation. This is because if the concentration of AFM1 is lower, then the amount of the conjugate attached with the antibody will become higher, and the enzyme from the conjugate will oxidize more TMB. This will generate a higher current. On the other hand, the higher the concentration of AFM1, the lower the attachment of conjugate with the antibody; the amount of TMB oxidized will be reduced, and due to this, the current generated will be lower. The measure of the quality of this particular kind of sensor can be found in its inhibiting efficiency: the higher the current drop at a certain concentration, the better the sensitivity of the device. 

In both functionalized and unfunctionalized electrodes, 0.01 µg/L to 1 µg/L of AFM1 was detected as the lowest and highest working ranges, respectively. As shown in Figure 5 above, dispense-printing and functionalization improved the sensitivity, since the inhibiting effect was higher than for the unfunctionalized electrode. SWCNTs helped to increase the surface area of the WE [1], which led to attachment of more specific antibodies, consequently improving the sensitivity of the electrode.

#### 3.2.2. Testing in Spiked Milk

The skimmed milk was then spiked with different concentrations of AFM1 to test it as a real sample. The results are shown in Figure 6. For the spiked milk, the immunosensor showed a similar trend as for buffer in the observed range. 

The limit of detection (LOD) in spiked milk were 0.055 and 0.0259 µg/L for unfunctionalized and functionalized sensors, respectively, while the working range was similar to that of the buffer. Compared to unfunctionalized, the SWCNT-functionalized sensor was more sensitive and its LOD was in line with the regulations. On the other hand, when the LOD for spiked milk was compared with that of the buffer solution, it showed slightly lower results. This may have been due to some interference between proteins or remaining fat globules and antibodies on the WE, and requires deeper investigation. 

## 4. Conclusions 

In this work, a flexible, dispense-printed electrochemical immunosensor was developed to analyze AFM1 in both buffer and spiked milk samples, using MAb for molecular recognition. This system allowed quantification of AFM1 in milk in an easy and cost-effective manner. The LOD of the sensor was 0.02 µg/L and 0.0259 µg/L, with a detection range of 0.01 to 1 µg/L for the buffer and milk samples, respectively. Both the sensitivity and detection range were in accordance with most of the existing limits imposed by law for milk and milk products. Therefore, the use of this sensor could help to safeguard consumers’ health, with the advantages of being inexpensive, rapid, and easy to operate, and thus suitable for application in milk collection points or on milk processing lines. In addition to this, the flexibility of dispense-printing allows the employment of almost any printing material—conductors, semiconductors, insulators—and the realization of arbitrary shapes, paving the way to custom-made and application-tailored biosensors. In future work, we will attempt to develop a much faster method using different approaches to improve the analysis and detection time.

## Figures and Tables

**Figure 1 sensors-19-03912-f001:**
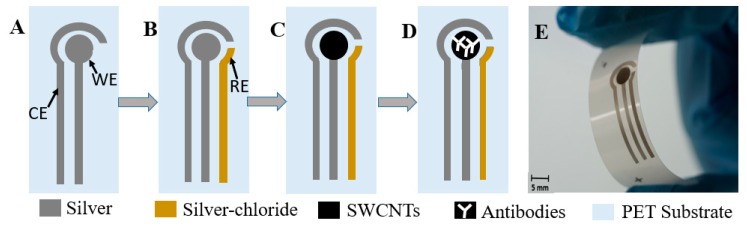
Schematic diagram of the fabricating process of the biosensors: (**A**) printing working electrode (WE) and counter electrode (CE), (**B**) printing WE with AgCl by alignment, (**C**) spray depositing single-walled carbon nanotubes (SWCNTs), (**D**) immobilization of antibody, and (**E**) final biosensor.

**Figure 2 sensors-19-03912-f002:**
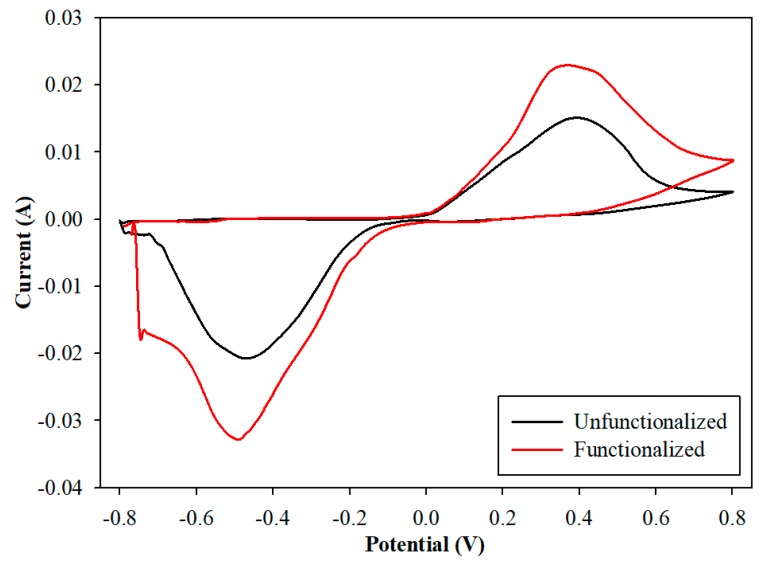
Cyclic voltammetry unfunctionalized vs. functionalized electrode in potassium hexacyanoferrate (III) solution.

**Figure 3 sensors-19-03912-f003:**
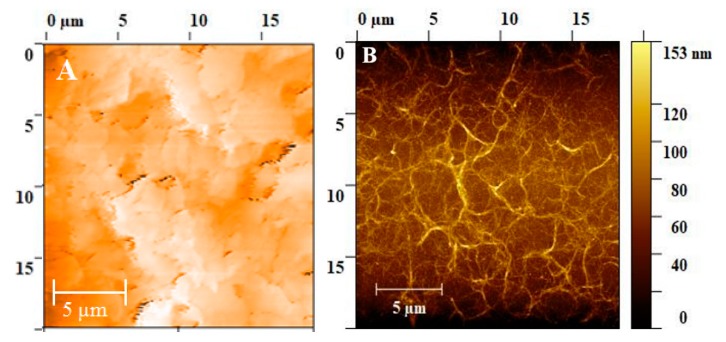
Atomic force micrographs (AFM) of the electrode: (**A**) unfunctionalized Ag electrode and (**B**) functionalized electrode with SWCNTs.

**Figure 4 sensors-19-03912-f004:**
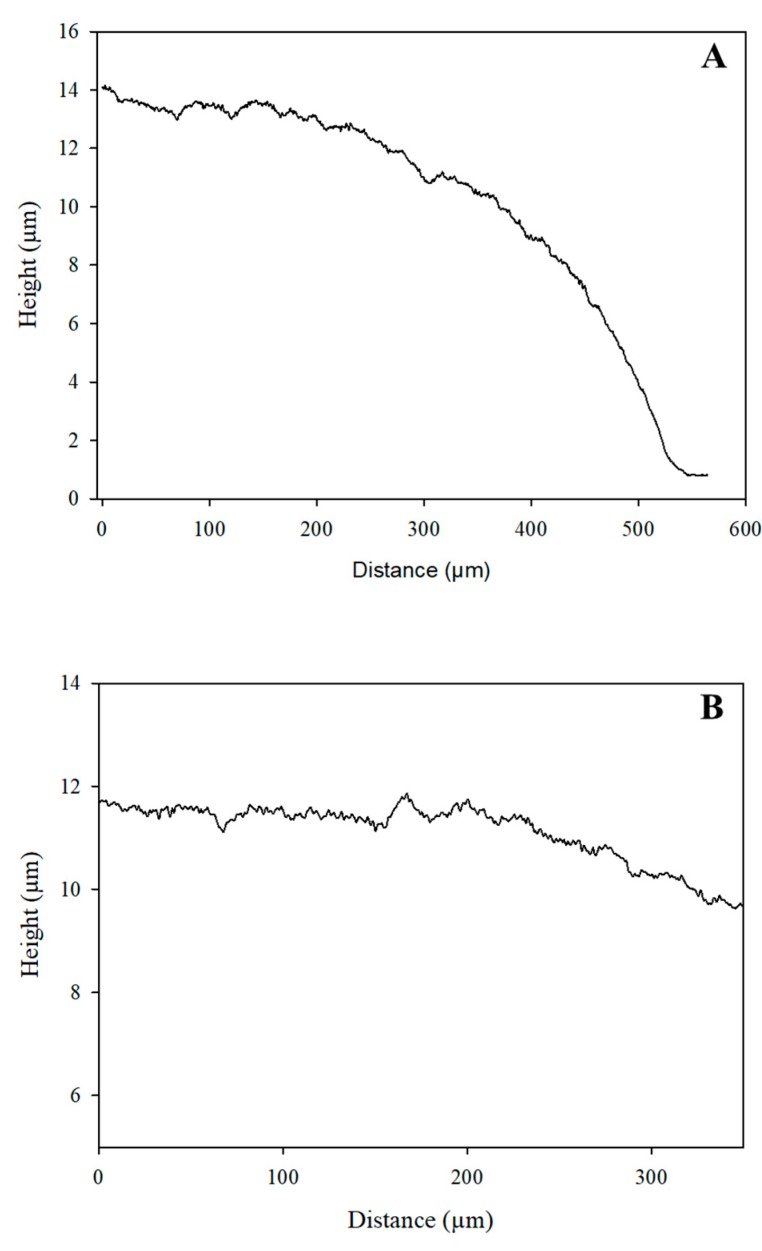
2D profile of the electrode: (**A**) step height and (**B**) surface roughness.

**Figure 5 sensors-19-03912-f005:**
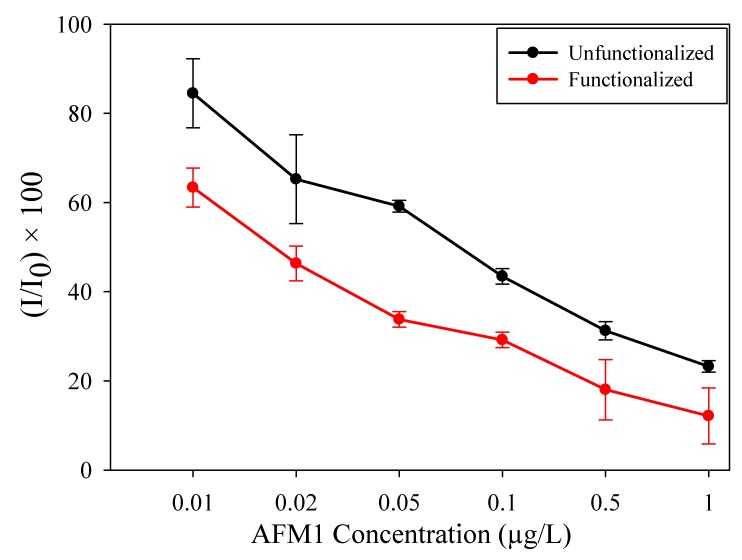
Chronoamperometric measurement of AFM1 in buffer solution; A_0_ is the sensor response without AFM1.

**Figure 6 sensors-19-03912-f006:**
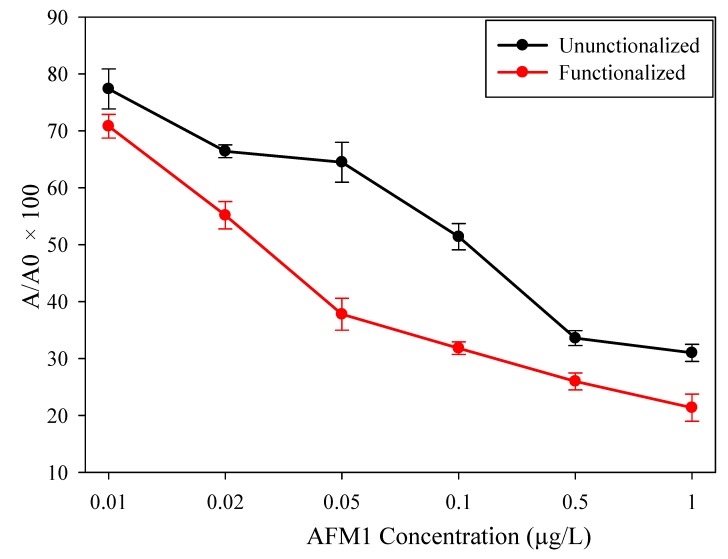
Chronoamperometric measurement of AFM1 in spiked milk sample; A_0_ is the sensor response without AFM1.

**Table 1 sensors-19-03912-t001:** Sensor response for AFM1 standards using phosphate-buffered saline (PBS) solution for unfunctionalized (without SWCNTs) and functionalized (with SWCNTs) sensors.

AFM1 Concentration (µg/L)	Unfunctionalized			Functionalized		
	Generated Current (µA)	Normalized Current (A/A_0_)	SD	Generated Current (µA)	Normalized Current (A/A_0_)	SD
0.00	−61.1	100.0	1.43 × 10^−5^	−92.1	100.0	1.8 × 10^−5^
0.01	−52.1	84.5	3.3 × 10^−5^	−63.2	63.4	1.05 × 10^−5^
0.02	−41.4	67.7	9.37 × 10^−6^	−46.1	46.4	1.08 × 10^−5^
0.05	−39.0	63.8	1.88 × 10^−5^	−33.8	33.8	8.14 × 10^−6^
0.1	−26.9	44.0	1.29 × 10^−5^	−30.2	30.2	1.28 × 10^−5^
0.5	−20.8	34.1	1.11 × 10^−5^	−16.2	18.0	5.47 × 10^−6^
1.0	−15.2	24.8	2.49 × 10^−5^	−13.2	12.1	2.31 × 10^−6^

**Table 2 sensors-19-03912-t002:** Sensor response for AFM1 standards using spiked milk sample for unfunctionalized (without SWCNTs) and functionalized (with SWCNTs) sensors.

AFM1 Concentration (µg/L)	Unfunctionalized			Functionalized		
	Generated Current (µA)	Normalized Current	STD	Generated Current (µA)	Normalized Current	STD
0.00	−93.2	100.0	3.52 × 10^−5^	−90.8	100.0	2.08 × 10^−5^
0.01	−72.1	78.3	1.13 × 10^−6^	−64.3	71.2	3.17 × 10^−5^
0.02	−61.9	66.5	9.83 × 10^−8^	−50.1	45.0	7.89 × 10^−6^
0.05	−60.1	64.5	4.47 × 10^−6^	−34.3	37.9	4.84 × 10^−5^
0.1	−47.9	51.4	1.18 × 10^−5^	−28.9	31.9	2.15 × 10^−5^
0.5	−31.3	33.6	4.81 × 10^−7^	−23.6	25.2	1.27 × 10^−6^
1.0	−28.9	31.0	6.65 × 10^−7^	−19.4	21.4	2.73 × 10^−5^

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
