# Peer review of "Development of Flexible Dispense-Printed Electrochemical Immunosensor for Aflatoxin M1 Detection in Milk"

_sensors, 2019, doi:10.3390/s19183912_

Round 1

Reviewer 1 Report

The Authors describe an electrochemical sensor functionalized with single wall carbon nanotubes for detection of mycotoxins containing in milk, and show its impact on detection limit. The motivation and aim of this research are clear and easy to understand, but there are some portions which cannot be understood due to the lack of explanation. This paper will be considered for publication in biosensors, after answering the comments shown below.

1.

L.54: "State-of-the-art of detection of AFM1 is carried out using classical methods like chromatography and enzyme-linked immunosorbent assay (ELISA) [19], which are time-consuming, require expensive equipment and high expertise in addition to sample pre-treatment (e.g. extraction, purification and detection and quantification) [11, 12, 20 , 21]."

L.153: "As a last step, for the competition of the analyte with the conjugate, 4 μL of PBS buffer or spiked milk sample, both containing AFM1 standards, was placed on top of the WE with 4 μL of the AFM1–HRP conjugate, obtained from an ELISA kit (1:10 dilution). Next, curing was performed at 37oC for 2 hr. The sensor was rinsed with 50 μL of PBS-T and kept at room temperature to dry. Then, a solution of 70 μL of 5 mM of TMB and 1mM H2O2 in citrate buffer, that contains 0.1 M KCl, was drop-casted on the three electrodes (CE, WE and RE) and kept for 25 minutes at room temperature."

àAlthough the Authors mention that classical methods are time-consuming, their developed method also seems to need a long time, for example, curing for 2h, and keeping drop-casted TMB for 25 min. Is it really fast method?

2.

L. 66: "However, it needs a mask to structure the biosensor, it consumes a considerably high amount of ink,…" 

L. 230: "The average step-height of the electrode was 13.75 μm while the surface roughness as shown in Figure 4B 0.9838 μm." 

àScreen printing can form not only thick but also thin patterns. For example, it has been reported that thickness of a screen-printed pattern was 4-5 μm [A]. Also, thickness of electrodes of the latest multilayered ceramic capacitor (MLCC) is said to be ~1 μm or less. In this point, the fabrication method using a dispenser does not seem to be superior to screen printing.

*A. Ito et al., Thin Solid Films, 516, 4613-4619 (2008).

3.

L.94: polyethylene (PET)--> polyethylene terephthalate (PET) 

Reviewer 2 Report

The following report shows the development of an immonumsensor, but it lacks a lot of information that could be considered important, for example, there is no clear response of the analyte before and after fusionalization. There is no single evidence to prove the presence of the biological component on the electrode surface. there are also no studies of stability or interference and the mattrix effect. The authors cite a very similar article that contains all these studies. Biosens Bioelectron. 2005, 15; 21 (4): 588-96. The calibration curves have a very small linear range. They do not indicate how the cell was prepared or if it was by direct drop on the electrode surface. The electrode surface characterization, I suggest should be done with the same technique before and after the modification and not with two different techniques before and after the modification. At least one of the amperograms must be placed next to the calibration curve. in table two they do not indicate the value detected in the milk sample nor the value after the enrichment with a standard of known concentration. in Characterization of electrical properties that indicates this value. It should be compared with other similar electrodes or compared with the value before modification to look at the range of change. I consider that this work must be completed before being published in sensors

Round 2

Reviewer 1 Report

The Authors appropriately answered my comments and revised their manuscript, but please consider one additional thing.

Although I understand that the proposed method is significantly fast compared to ELISA, "less than 10 min" seems still long in practical use. If so, I suggest to add a sentence explaining that the Authors will attempt to develop a much faster method as a next step.

Author Response

Thank you for your valuable comment. Based on your suggestion, we added the following sentence in the conclusion part as future work. 

"In future work, we will attempt to develop an even faster method using different approaches to improve the analysis and detection time."

Reviewer 2 Report

the authors have considered all the suggestions,
but in a recent literature search I found a very similar work published in 2019 IEEE International Conference on Flexible and Printable Sensors and Systems (FLEPS). DOI: 10.1109/FLEPS.2019.8792320
of the same authors. As I could not access the document, I believe that it is necessary for the authors to explain what the differences are with this published work and with the new one that is intended to be published in sensors. because it is not the summary of a congress it is a publication of a special edition.  

Author Response

Thank you for your comment. We have been working on AFM1 detection using a different approach. The published work that you have mentioned and this submitted manuscript are different works. This paper is not an extension of the conference paper.  The work that has been published on FLEPS 2019 is focused on bare electrode pretreatment using NaOH and Oxygen Plasma to remove the insulating polymers added to the printing ink to enhance its adhesion to the substrate. While in this work we used nanotubes (SWCNTs) which are spray deposited on top of the electrode to increase the surface area to volume ratio which can help to increase the amount of antibody attachment to the electrode and consequently improves the sensitivity and limit of detection.